# Do contemporary (1980–2015) emissions determine the elemental carbon deposition trend at Holtedahlfonna glacier, Svalbard?

Meri M. Ruppel[1], Joana Soares[2], Jean-Charles Gallet[3], Elisabeth Isaksson[3], Tõnu Martma[4], Jonas Svensson[1,2], Jack Kohler[3], Christina A. Pedersen[3], Sirkku Manninen[1], Atte Korhola[1], and Johan Ström[5]

[1]Department of Environmental Sciences, University of Helsinki, Helsinki, FI-00790, Finland

[2]Finnish Meteorological Institute (FMI), Helsinki, FI-00560, Finland

[3]Norwegian Polar Institute, Tromsø, NO-9296, Norway

[4]Department of Geology, Tallinn University of Technology, Tallinn, EE-19086, Estonia

[5]Department of Environmental Science and Analytical Chemistry ACES (Atmospheric Science Unit), Stockholm University, Stockholm, SE-11418, Sweden

*Correspondence to*: M. M. Ruppel (meri.ruppel@helsinki.fi)

## Abstract

The climate impact of black carbon (BC) is notably amplified in the Arctic by its deposition that causes albedo decrease and subsequent earlier snow and ice spring melt. To comprehensively assess the climate impact of BC in the Arctic, information on both atmospheric BC concentrations and deposition are essential. Currently, Arctic BC deposition data are very scarce, while atmospheric BC concentrations have been shown to generally decrease since the 1990s. However, a 300-year Svalbard ice core showed a distinct increase in EC (elemental carbon, proxy for BC) deposition from 1970 to 2004 contradicting atmospheric measurements and modelling studies. Here, our objective was to decipher whether this increase has continued in the 21st century, and to investigate the drivers of the observed EC deposition trends. For this, a shallow firn core was collected from the same Svalbard glacier, and a regional-to-meso-scale chemical transport model (SILAM) was run from 1980 to 2015. The ice and firn core data indicate peaking EC deposition values at the end of the 1990s, and lower values thereafter. The modelled BC deposition results generally support the observed glacier EC variations. However, the ice and firn core results clearly deviate from both measured and modelled atmospheric BC concentration trends, and the modelled BC deposition trend shows variations seemingly independent from BC emission or atmospheric BC concentration trends. Furthermore, according to the model ca. 99 % BC mass is wet-deposited at this Svalbard glacier, indicating that meteorological processes such as precipitation and scavenging efficiency have most likely a stronger influence on the BC deposition trend than BC emission or atmospheric concentration trends. BC emission source sectors contribute differently to the modelled atmospheric BC concentrations and BC deposition, which further supports our conclusion that different processes affect atmospheric BC concentration and deposition trends. Consequently, Arctic BC deposition trends should not directly be inferred based on atmospheric BC measurements, and more observational BC deposition data are required to assess the climate impact of BC in Arctic snow.

## 1. Introduction

Black Carbon (BC) is a carbonaceous fine particle with strong light-absorbing ability. It is produced by natural and anthropogenic incomplete combustion of biomass and fossil fuels, and may be transported with prevailing winds over thousands of kilometres from its emission sources (e.g. Ramanathan and Carmichael, 2008; Bond et al., 2013). It poses a global environmental threat by warming the atmosphere, but the climate impacts of BC are amplified in the Arctic where its deposition on snow and ice decreases surface reflectance, hastens snow and ice melt which further decreases the reflectivity (e.g. Hansen and Nazarenko, 2004). Globally, BC is the second most important climate warming agent after carbon dioxide, but in the Arctic it may be due to the snow-albedo-feedback even more important to the observed warming and melting than greenhouse gases (e.g. Flanner et al., 2007, 2009; Bond et al., 2013).

Atmospheric BC concentrations have been monitored in the Arctic starting since 1989 at Alert (Canada), Barrow (USA) and later at Zeppelin (Ny Ålesund, Norway), and the observations show a 40 % decrease from 1990 to 2009 (Sharma et al., 2013). Furthermore, measurements from northern Finland showed a 78 % decrease in atmospheric BC concentrations between 1971 and 2011 (Dutkiewicz et al., 2014). This observed decrease is mostly attributed to the fall of the USSR and a resulting decrease in BC emissions in major source areas of Arctic BC (e.g. Sharma et al., 2013). However, atmospheric observations reflect the effect of BC on Arctic climate only partially, as the climate effect of BC deposited on high-reflectance snow and ice surfaces is notably stronger than of atmospheric BC (Flanner et al., 2007, 2009; Bond et al., 2013). As 85–90 % of BC is suggested to be wet-deposited in the Arctic (Wang et al., 2011), and the BC proportion bound by precipitation is mostly not recorded by atmospheric measurements, the BC emission and atmospheric BC concentration trends may not reliably represent the BC deposition trend. Therefore, to comprehensively assess the effects of BC in Arctic climate change, also observations on its deposition rate and trend in the area are essential.

Ice cores represent a valuable means to study BC deposition as they accumulate direct evidence of contaminant deposition in chronological order, potentially for hundreds to thousands of years. Ice core records are irreplaceable when evaluating e.g. contemporary atmospheric or snow BC concentration variations in the context of past BC variations, and when evaluating the role of these variations for the observed climate change in the Arctic and beyond. Despite the importance of ice core records in deciphering the role of BC in Arctic climate change, relatively few records exist at present. Four continuous BC ice core records covering ca. 1750 to 2013 have been published from Greenland (McConnell et al., 2007; McConnell and Edwards, 2008; McConnell, 2010; Keegan et al., 2014) and one 300-year record (1700 to 2004) from Svalbard (Ruppel et al., 2014). The high-elevation Greenland records indicate a BC deposition peak around 1910 followed by rapidly decreasing deposition until 1950, and more or less stable, almost preindustrial values until the present (McConnell, 2010). The Svalbard ice core clearly concurs with the Greenland records for the early 20th century, but unexpectedly shows a pronounced increase in BC concentrations and deposition from 1970 to the top of the core, 2004 (Ruppel et al., 2014). The reasons for the observed post-1970 BC deposition increase in Svalbard, while at the same time Greenland ice cores, atmospheric measurements (e.g. Sharma et al., 2013) and model results (e.g. Koch et al., 2011) suggest decreasing BC values, was left partly unresolved (Ruppel et al., 2014). Increasing flaring emissions from northern Russia in the Barents Sea area that do not reach the Greenland ice coring sites due to restricted isentropic uplift in the Arctic, and potentially increasing wet-scavenging efficiency due to increasing temperatures particularly around Svalbard were the leading hypotheses (Ruppel et al., 2014). A similar rapid increase in BC fluxes between ca. 1970 and 2013 was also observed in two lake sediment records from northern Finland (Ruppel et al., 2015).

The increasing BC deposition on the Svalbard glacier has significant effects on the radiative budget of this site and concurs with substantially increased summer melting of the glacier since the 1980s (Ruppel et al., 2014). The increased melt of the glacier is better explained by the combination of observed increasing summer temperatures and the increasing BC concentrations, than by increasing temperatures alone. To estimate the extent of the climatic implications suggested in Ruppel et al. (2014) it is essential to solve whether the observed increasing BC deposition trend in Svalbard since 1970 can be corroborated with other data. Furthermore, it is necessary to thoroughly assess the BC sources responsible and the deposition processes associated to the observed increase, as these may affect also other parts of the Arctic.

Here, our objective is to resolve what the BC deposition trend has been during the last 10 years on the previously studied Svalbard glacier, Holtedahlfonna. For this, a new 14.7 m deep firn core and a 2.5 m thick accumulated winter-spring snow profile were collected from Holtedahlfonna in April 2015. BC was analysed from the samples as elemental carbon (EC) with the same methodology (thermal-optical with the Eusaar 2 protocol) as in Ruppel et al. (2014). In addition, the source of the BC deposited at the glacier is investigated using a meso-to-global scale chemical transport model SILAM (Sofiev et al., 2012). The results may have significant implications for the comprehensive assessment of the impact of BC in the recent past, present and future Arctic climate system.

## 2. Material and methods

### 2.1. Site description, and field sampling

Svalbard is an archipelago located in the Arctic Ocean (Fig. 1). It has relatively mild climate despite its location at high latitudes due to an intrusion of the North Atlantic current bordering western Svalbard and its location in the pathway of both Arctic and North Atlantic cyclones. Svalbard is covered to 60 % by glaciers of which the majority has retreated during the last 15-40 years (Nuth et al., 2010), and even the glaciers situated at highest elevation (ca. 1200 m a.s.l.) experience frequent surface melt in the summer (e.g. Beaudon et al., 2013).

Holtedahlfonna is a ca. 300 km$^2$ ice field in western Svalbard, located 40 km northeast of the Ny Ålesund research station (Fig. 1). A 125 m deep ice core was drilled in April 2005 at 1150 m elevation, at a saddle point of a mountain ridge at the edge of Holtedahlfonna (Fig. 1, N 79'08.15, E 13'16.20). This coring location was selected based on the current knowledge of the subglacial bedrock topography. The ice core was estimated to cover ca. 300 years, and elemental carbon (EC) was analysed from the inner core section (Ruppel et al., 2014). The EC analysis revealed an unexpected increase in the Arctic from 1970 to 2004 (i.e. the top of the core). To confirm the observed trend and to decipher whether the trend had continued after 2004, a new 14.7 m deep firn core was collected on April 19th 2015 from the same glacier, at 1120 m elevation (N 79'08.424, E 13'23.639). The new coring site was located ca. 2.8 km southeast from the 2005 coring site (Fig. 1), in the vicinity of a mass balance measurement stake, and thus annual snow accumulation measurements are available from this site since 2003. To Before drilling, the top 80 cm of the snow pack were removed to obtain a hard surface to drill 80 cm of the surface snow pack were removed. The core was collected with a PICO drill, and a depth of 14.7 m below the snow surface was reached. A combination of time constraint and type of drill did not permit drilling deeper. The firn was retrieved in ca. 60–100 cm sections and was immediately packed into labelled plastic bags.

In addition, a 2.53 m deep snow pit was dug on April 21$^{st}$ ca. 15 meters away from the firn coring site, to study the top snow layers missed by the firn core, and the annual EC variation in more detail. The snow pit was dug down to the

previous summer surface that was identified both by a (the only) hard layer in the snow pack, and the snow accumulation data retrieved from the mass balance measurements at the site. The snow column thus covers the accumulated snow from the end of summer 2014 to April 2015. The snow was first excavated and snow physical properties were recorded, particularly the snow stratigraphy, which allowed the identification of different layering in the snow pack. In total, 13 samples were collected into whirl-pack bags, resulting in on average a 20 cm vertical resolution for EC measurements in snow. The firn core and snow samples were stored frozen and transported to the Norwegian Polar Institute (NPI), Tromsø, Norway.

The firn core was cut in a freezer laboratory (-22 °C) using a thin blade band saw. Each vertical ice-core section was split to sub-samples assigned to oxygen and hydrogen isotope ratio, and EC analyses. The outer 1-2 cm layer of the firn core were removed and the isotope and EC samples were cut from the inner part of the firn core protected from possible contamination during drilling, packing and handling until distributed into clean sample vials. As the EC concentration of the firn samples was expected to be quite low for the thermal-optical method to detect, the sample sizes for EC measurements were kept relatively large. The core was divided into 14 vertical sub-samples for EC measurements of ca. 1 m total length each, and an average horizontal cross section of 4.2 by 4.5 cm, equal to an average ~~surface area~~ cross sectional area of 19 cm$^2$ ($\pm$ 2 cm$^2$). The sub-samples generated between 0.9 and 1.2 L melt water.

## 2.2. Filtering and EC analysis

To ensure comparability, the filtering and EC analysis of the firn core and snow samples were performed in the same facilities and with same instruments as in Ruppel et al. (2014), which followed the original procedure of Forsström et al. (2009). The frozen samples were melted and immediately filtered through pre-combusted (at 800 °C for 4 hours) quartz fibre filters (Munktell) in a glass filtering system. All parts of the filtering system were cleaned between each sample using distilled water and a brush. Blanks (5 samples) were prepared to check for possible contamination in the filtering system. The filters were allowed to dry in individual petri dish containers in a clean cupboard and subsequently individually wrapped into aluminium foil and stored in a refrigerator (+6–8 °C) before analysis.

The filters were analysed for EC using a thermal optical method (TO, Sunset Laboratory Inc., Forest Grove, USA; Birch and Cary, 1996) with the latest recommended thermal sequence EUSAAR_2 (Cavalli et al., 2010) at Stockholm University. The method separates in the first stage organic and carbonate carbon (OC and CC, respectively) from the filters under increasing temperature steps in a helium atmosphere, while EC evolves from the filters in the second stage under a helium-oxygen atmosphere at temperatures reaching 850 °C. During analysis, the transmittance of the filter is monitored using laser light, which allows for optical correction of charring, i.e. potential pyrolysis of OC to EC during the analysis (Cavalli et al., 2010). All blanks showed EC concentrations well below detection limit of the analysis method (0.2 EC µg cm$^{-2}$). ~~A more detailed description is given in Ruppel et al. (2014).~~

The used methodology includes uncertainties that are described in more detail in Ruppel et al. (2014). In short, in liquid samples (i.e. melted snow and ice) smallest EC particles may go through the filter (e.g. Torres et al., 2014) leading to a quantified undercatchment of ca. 22 % for the used filtering set-up (Forsström et al., 2013). In addition, from each filter sample (11.34 cm$^{-2}$) only a small punch (1.5 cm$^{-2}$) is analysed for EC. To evaluate the uncertainties caused by this subsampling, triplicate analyses were prepared for five ice core samples. These measurements (Fig. 4c) showed an average relative standard deviation of 8.5 % (range of relative standard deviation = 5.3–13.7 %) that is smaller than reported e.g. in Ruppel et al. (2014; 19.6 % average relative standard deviation). Combined, i.e. added together in quadrature, these error sources cause a ca. 23.6 % uncertainty in our current EC measurements.

## 2.3. Dating of firn core

The 14.7 m firn core was dated by preparing a composite estimate based on annual layer counting using the seasonal variability in the oxygen ($\delta^{18}$O) and hydrogen ($\delta^2$H) stratigraphy in combination with snow density, and mass-balance measurements (stake 10) next to the coring site recorded since 2003. For the oxygen and hydrogen isotope analysis the core was sampled at 5-cm vertical resolution following methods described in Divine et al. (2011). The isotope analyses were performed at Tallinn University of Technology using a Picarro L2120-I water isotope analyser with a high precision AO211 vaporiser. The results were calibrated to V-SMOW scale. Reproducibility of the $\delta^{18}$O and $\delta^2$H measurements was ±0.1 and ±1‰, respectively.

There are very pronounced variations with large seasonal amplitudes in the water isotope records in the uppermost two meters of the core assumed to be due to different atmospheric sources of precipitate. These annual variations gradually get smoothed out due to diffusion during the firnification process (Fig. 2), rendering the distinction between years more difficult with increasing depth. Therefore, supporting data from the mass balance stake are useful for dating. Stake 10 has been visited and maintained regularly since 2002 and thus an annual mass balance of the study site is available from 2003. By combining the density and depth data from the firn core, the snow water equivalent along the core profile could be obtained, and was compared with the mass-balance (snow water equivalent accumulation) data measured at the stake since 2003. Thereby, the limit of years (September measurement points at the stake) could be determined as a function of depth, and subsequently the core could be dated. The density and water isotope inferred dating of the core match well with the mass balance inferred limits of years (Fig. 2).

## 2.4 Atmospheric modelling

The System for Integrated modeLling of Atmospheric composition Model, SILAM (Sofiev et al. 2008), a model developed by the Finnish Meteorological Institute was run for a simulation between 1980 and 2015 to study BC deposition variations and the contribution of different sources of BC deposited at Holtedahlfonna. SILAM is a meso-to-global scale chemical transport model. For this study, the model was driven by ERA-Interim (Dee et al., 2011) meteorology, and by global MACCity anthropogenic BC emissions (Granier et al., 2011) updated with ECLIPSE emission dataset for flaring (Klimont et al., 2013), and natural fire (open biomass burning) emissions (Lamarque et al., 2010) shown in Figure 3. Generally, BC emissions north of 40° N are considered significant for the Arctic (AMAP, 2011). Unfortunately, to our knowledge, there is no single continuous data source of natural fire BC emissions from 1980 to 2015. The Lamarque et al. (2010) dataset we used here is based on estimated burned land area, and extends until the year 2008. Between 2008 and 2015 constant emissions of 2008 were used. Kaiser et al. (2012) and Soares et al. (2015) have shown that the emission data estimated by burned area data tend to be underestimated. An alternative data set of natural fire BC emissions, that is considered more reliable, is based on satellite images, IS4FIIRES (Soares et al., 2015). However, the satellite emission data are available only since 2003 and show considerably higher values, which would cause a step change in the emissions if the data sets were combined. As our objective is to examine trends for 1980 to 2015, it is more reasonable to use the longer Lamarque et al. (2010) data set in our simulations. The total global BC emissions have increased in the study period while north of 40° N they have decreased (Fig. 3). Svalbard receives atmospheric transportation dominantly from Eurasia (e.g. AMAP, 2011), and anthropogenic BC emissions from this region have decreased in the study period while natural fire emissions have increased (e.g. Bond et al., 2007; Lamarque et al., 2010).

The model was run through the period between 1980 and 2015 with 1-hour temporal resolution, $0.72° \times 0.72°$ horizontal resolution, and 29 Hybrid sigma-pressure vertical levels. This investigated time period was constrained by the availability of meteorological ERA-Interim data. The dispersion model considers BC as an inert pollutant, with size distribution described by a single bin with size ranging from 0.001 to 1 μm in dry ~~Dp~~particle diameter (*Dp*). Production was integrated over each size bin while dry and wet removal rates were calculated using mass-weighted mean diameter in each bin. Depending on particle size, which takes into account the particle hygroscopic growth, mechanisms of dry deposition varied from primarily turbulent diffusion driven removal of fine aerosols to primarily gravitational settling of coarse particles (Kouznetsov and Sofiev, 2012). Wet deposition distinguished between sub- and in-cloud scavenging by both rain and snow (Horn et al., 1987; Smith and Clark, 1989; Jylhä, 1991; Sofiev et al., 2006). The sources for BC deposited at Holtedahlfonna were investigated by tagging the different emission sectors while computing atmospheric dispersion of BC. Subsequent to the SILAM runs, a multilinear regression model based on the median values for atmospheric BC concentrations and deposition for every single year, between 1980 and 2015, was used to estimate the slope of the modelled temporal BC trends, with coefficients being estimated with 95 % confidence intervals. An F-test was applied to test if the linear regression relationship between the response and predictor variables was significant.

Generally, ~~l~~like many models, SILAM agrees better with observations closer to sources than in the Arctic. In the Arctic the modelled BC ~~levels~~ concentrations and deposition are systematically low, but the seasonality in atmospheric BC concentrations is captured ~~well~~, specifically capturing the Arctic Haze period (see Fig. 5 and discussion below).

## 3. Results

### 3.1. Snow pit EC data

The EC variations in the snow pit covering the end of summer 2014 to April 2015 are shown in Figure 4a. The EC concentrations ranged between 4.7 and 20.3 μg L⁻¹ which are ~~generally similar~~in the same range ~~as~~to EC concentrations of 1.4, 9.4 and 11.6 μg L⁻¹ previously measured at the same site in spring surface snow of 2007, 2008 and 2009, respectively (Forsström et al., 2009, 2013; Fig. 4b). The snow pit samples show a similar seasonal trend in EC concentrations as previously observed in Arctic snow packs with elevated concentrations during spring and summer and lower values in the autumn and winter (Doherty et al., 2010, 2013).

### 3.2. Firn core EC data

The EC concentrations of the shallow firn core are between 3.5 and 24.6 μg L⁻¹ with an average of 10.4 μg L⁻¹ (Fig. 4b). The firn core EC concentrations match the snow pit EC observations for the overlapping part from 80 to 253 cm from the snow surface (Fig. 4b). The annual deposition of EC to the firn core was calculated using the dating (section 2.3.) of the core. The EC deposition values in the firn core range from 2.8 to 19 mg m⁻² yr⁻¹ (on average 10 mg m⁻² yr⁻¹). Table 1 presents annual (averaged over calendar years) EC concentrations and deposition for 2006 to 2014.

Due to the comparably low temporal resolution of the EC samples no annual variation can be detected in the firn core, although the observed EC variation may be partly caused by some samples covering more of the high BC laden spring to summer snow (e.g. two vs. zero spring layers) compared to cleaner winter snow (cf. Ruppel et al., 2014). The firn core is too short to indicate any clear temporal trend, but in general, the EC concentrations and deposition seem to be on a lower

level from 2005 to 2011 and to increase to higher levels from 2012 to 2015 (Fig. 4c and d, Table 1). The temporal trend of EC deposition is similar to the EC concentration trend observed in the core (Fig. 4c and d).

### 3.3. Modelled BC data

To evaluate the performance of SILAM for Svalbard and BC, atmospheric BC observations made at the Zeppelin (Ny-Ålesund) monitoring site were compared to model results from the correspondent model grid-cell in Figure 5. Figure 5a shows the model results from 1980 to 2015 while atmospheric observations were available only for 2002 to 2011. Both the observations and model results show large variation in atmospheric BC concentrations from one year to the next, but with an overall decreasing trend (Fig. 5a). However, compared to the observations, the model significantly underestimates the atmospheric BC concentrations (on average by a factor of five). Such underestimations of atmospheric BC concentrations are particularly common for the Arctic where previous comparisons to observations have shown atmospheric BC concentrations being underestimated in chemistry models by up to a magnitude (e.g. Koch et al., 2009; Lee et al., 2013; Dutkiewicz et al., 2014).

Figures 5b and c present the seasonality of observed and modelled monthly BC concentrations for 2006 and 2007. The model captures the seasonality seen in the observations but fails to reproduce the magnitudes observed especially in spring time. Note that the timing of observed spring peaks (Arctic Haze) varies from year to year. This corroborates with several multi-model studies (Shindell et al., 2008; Koch et al., 2009; Eckhardt et al., 2015) showing that atmospheric models are usually not able to simulate the seasonality of BC in the Arctic precisely, typically underestimating the Arctic Haze season occurring during the winter and early spring. A more detailed discussion on the uncertainties of the model and input driving the runs is presented in Section 4.

The results of modelled atmospheric BC concentrations and BC deposition at Holtedahlfonna are presented in Figure ~~5~~6. The modelled annual atmospheric BC concentrations decrease quite constantly from 1990 onwards after notably higher values modelled for the 1980s (slope $-1.3 \times 10^{-5}$ $\mu$g m$^{-3}$ yr$^{-1}$; $p < 0.001$). The modelled BC deposition on the other hand shows significant variation from year to year with no clear ~~decadal~~ trend over the study period. Statistically, the deposition trend decreases weakly over 1980 to 2015, but this trend is not significant (slope = $-3.9 \times 10^{-3}$ $\mu$g m$^{-3}$ yr$^{-1}$; $p = 0.09$). The modelled atmospheric BC concentration and deposition trend correlate only weakly ($r = 0.29$, p = 0.08) over the study period.

The model results suggest that the total BC deposition is dominated by ~~98.7 % of BC is~~ wet ~~deposited~~ deposition at Holtedahlfonna (98.7 %).

There are notable differences in the source contributions for the modelled BC deposition and atmospheric BC concentrations at Holtedahlfonna (Fig. ~~6~~7). Over the period of 1980 to 2015 transport and domestic emissions are the most important sources for BC deposited at Holtedahlfonna (Fig. ~~6a~~7a), both with ca. 30 % contribution, while the domestic sector (43 % on average) is the most important emission source for atmospheric BC concentrations at the glacier, followed by the industry and transport sectors (Fig. ~~6b~~7b). For both the modelled atmospheric BC concentrations and deposition the contribution of domestic emissions has decreased during the investigated time period while the contribution of transport, including shipping, and natural fires has increased, and the contribution of industry and other sectors has stayed quite constant.

## 4. Discussion

### 4.1. Comparison of the snow and firn core EC data with the 2005 ice core

Previous EC concentrations from surface snow in 2007, 2008 and 2009 (Forsström et al., 2013) and the snow pit and firn core data collected from the Holtedahlfonna 2015 coring site (Stake 10), corroborate each other (Fig. 4b). However, the firn core EC concentrations measured at Stake 10 (an average of 10.4 µg L$^{-1}$) are notably lower than recorded in the 300-year ice core collected from a different site on the same glacier in 2005 (on average 35.8 µg L$^{-1}$) (Ruppel et al., 2014). On the other hand, the overall annual EC deposition in the firn core (on average 10 mg m$^{-2}$ yr$^{-1}$) compares quite well to the EC deposition recorded in the 300-year ice core (on average 11.2 mg m$^{-2}$ yr$^{-1}$). Yet, there is a notable drop of a factor of 2.5 in the EC deposition values from the last data point in the 300-year ice core (of 23.7 mg m$^{-2}$ yr$^{-1}$ deposited in the sample covering ca. 2001 to 2003), to the first sample in the firn core (9.3 mg m$^{-2}$ yr$^{-1}$ deposited ca. between mid-2005 to early-2006) (Fig. ~~7~~8). Regrettably, the new firn and old ice core do not temporally overlap, and therefore it cannot be confirmed whether the measurements at the separate coring locations are directly comparable. In the following we discuss the hypothesis of an actual rapid drop in EC deposition having occurred between the end of 2003 and mid-2005 at Holtedahlfonna, as suggested by the current data. Secondly, we explore the hypothesis that this difference in EC deposition is caused by local post-depositional factors at the coring sites, impeding the comparison of the cores. In addition, the sources for the deposited EC are examined.

### 4.1.1. Post-depositional processes affecting EC deposition at the two Holtedahlfonna coring sites

EC *concentrations* in snow and ice are strongly affected by numerous additional factors to atmospheric BC concentrations, such as EC scavenging efficiencies, precipitation amounts, and post-depositional processes of wind drift, sublimation and melt, that may dilute or concentrate EC in the snow (e.g. Doherty et al., 2010, 2013). Snow accumulation rates and post-depositional factors may vary locally, potentially causing differences in EC concentrations between the two Holtedahlfonna coring sites located 2.8 km apart (Fig. 1). Previous results on EC or BC concentrations in surface snow and full vertical snow profiles have shown that EC concentrations and column loads can vary substantially (commonly of a factor of two, but even a factor of 16 has been reported) even on a meter to meter scale due to post-depositional processes (e.g. Doherty et al., 2010, 2016; Svensson et al., 2013; Forsström et al., 2013; Delaney et al., 2015). Consequently, the similarity in EC concentrations measured at Stake 10 (2015 coring site) from the surface snow, snow pack and firn core samples gives confidence on the reproducibility of the used EC analysis method. At the same time, in light of the commonly found small scale horizontal variation in EC and BC concentrations discussed above, the differences in EC concentrations observed between the 2005 and 2015 Holtedahlfonna coring sites are not unexpected.

On the other hand, EC *deposition* at a specific site is generally not affected by post-depositional processes, as long as EC is not transported laterally after deposition (e.g. Ruppel et al., 2014). The atmospheric processes are expected to be the same for the Holtedahlfonna sites which consequently likely receive the same amount of EC input from the atmosphere. However, geomorphological properties of the coring sites, such as topography, differ between the sites, which may result in different amounts of snow being deposited at the sites by lateral transport (redistribution) of snow. The 2005 site is located 50 m higher in altitude at a point more exposed to wind activity than the 2015 coring site that is located on the central line of the glacier where mass balance measurements are performed (Fig. 1). Indeed, the 2005 core site records notably less annual net accumulated material (on average 0.5 m water equivalent per year (m w e yr$^{-1}$) between 1960 and 2004) than the 2015 site (on average 0.78 m w e yr$^{-1}$ between 2003 and 2015). The different snow accumulation rates may

indicate that different post-depositional processes affect or dominate at the sites, consequently causing the annual EC concentrations and deposition to diverge at the sites. The snow accumulation rate difference has significant consequences for the annual EC concentrations observed at the sites (higher concentrations at the 2005 site) if the same amount of precipitation is assumed for the sites (discussed below). However, if the precipitation amount at the sites is indeed the same, EC deposition is not directly affected by the snow accumulation rates (Ruppel et al., 2014), and additional factors are needed to explain the different EC deposition rates at the sites. The only processes by which the observed EC deposition in snow and/or an ice core could conceivably be notably higher at one of nearby locations receiving same precipitation and atmospheric EC deposition, is additional lateral or vertical transport of already deposited EC by wind activity. These processes affecting the snow accumulation, EC concentrations and deposition at the sites are discussed in more detail in the following.

The annual snow accumulation rate is the sum of snow accumulating (precipitation, wind drift) and reducing (ablation, run-off) processes. The precipitation amount at the sites is considered the same, and therefore wind drift, summer melt and sublimation are the probable causes for the different net snow accumulation at the sites. Summer melt occurs frequently on Holtedahlfonna (Beaudon et al., 2013). However, BC has a low post-depositional scavenging efficiency due to its hydrophobic properties, i.e. it is concentrated in melting snow and not flushed unless the melting is strong (e.g. Doherty et al., 2013). No summer surface run-off or high amounts of refrozen water (signalling strong vertical movement of melt water) in the snow stratigraphical record have been observed on Holtedahfonna, indicating that the summer melt on Holtedahlfonna is not strong enough to flush EC laterally or vertically. Therefore, it is unlikely that melting or run-off would cause the different EC deposition amounts at the two coring sites.

Consequently, wind drift and sublimation may be the most plausible post-depositional explanations for the observed differences in snow accumulation rate and EC deposition levels at the two coring sites, as these processes actually have the potential to remove or add snow and EC to the annual snow pack. Redistribution of snow mass by wind drift has significant impacts on the snow accumulation rates on Svalbard (Jaedicke and Gauer, 2005; Beaudon et al., 2011; Sauter et al., 2013). Sauter et al. (2013) showed that on Vestfonna ice cap, eastern Svalbard, up to 20 % of primary accumulated snow is redistributed by wind drift. To explain the higher EC deposition at the 2005 site it should receive more EC-laden snow by wind drift than the 2015 site. Higher wind drift could also explain higher EC concentrations at the site, since part of snow mass is sublimated during its transport (Sauter et al., 2013) which concentrates EC in ~~wind blown~~wind-blown snow. However, if the higher EC deposition at the 2005 site would be solely explained by it receiving more snow by wind drift than the 2015 site, then its snow accumulation rate should also be higher than that of the 2015 site. As the snow accumulation rate is actually lower at the 2005 than the 2015 site, wind drift cannot explain the differences alone.

Sublimation, which is a function of air temperature, humidity and wind speed, may affect the varying net snow accumulation rate at the Holtedahlfonna coring sites, as Arctic winter sublimation commonly reaches values of 10–50 % of total winter precipitation (Liston and Sturm, 2004 and references therein). During sublimation water is lost from the snow pack while EC is left behind and concentrated (e.g. Doherty et al., 2010, 2013). The 2005 site is most likely windier than the 2015 site, and may therefore be more prone to sublimation, which would result in the lower net snow accumulation rate observed at this site compared to the 2015 site. However, although significant amounts of water may be lost from snow/glacier surfaces due to sublimation, this process does not affect EC deposition rates.

Thus, to explain simultaneously the differences in snow accumulation rates and EC deposition amounts at the two Holtedahlfonna coring sites by post-depositional processes, a combination of high snow drift and sublimation would need

to be considered. However, the differences in snow accumulation rates and EC values between the sites are so large that based on current knowledge on the amount of snow remobilisation by wind and sublimation discussed above, it seems improbable that these processes would explain the differences between the sites alone.

Consequently, while the post-depositional processes certainly affect the measured snow accumulation rate and EC
concentration, and wind drift the EC deposition, none of these processes are neither alone nor together likely to entirely account for the different level of EC deposition observed in the 2005 and 2015 firn/ice cores. It is therefore more plausible that a drop in EC deposition has occurred between 2003 and 2005 at the Holtedahlfonna glacier. The magnitude of this drop remains uncertain, since the differences between the ice and firn core are affected by the above described post-depositional processes to an unknown extent. Sudden drops are not unprecedented in the 300-year Holtedahlfonna
record where EC deposition has dropped strongly, for instance, from peak values of 34 mg m$^{-2}$ yr$^{-1}$ around 1908 to 14 mg m$^{-2}$ yr$^{-1}$ in 1913 (Fig. 7̶8). At the same time, it should be kept in mind that previous long-term records comparing EC (or BC) variations at *different* coring locations on the same glacier analysed with the same methods are largely missing. Therefore, it is ultimately unverified whether such notable differences in EC values are common due to local differences between coring sites, or indicate actual EC variation events on the glacier.

*4.1.2. Variation in modelled atmospheric BC deposition at Holtedahlfonna between 1980 and 2015*

Atmospheric BC deposition at Holtedahlfonna (as at remote Arctic regions in general) is a complex end result of BC emissions within and outside of the Arctic, the prevailing atmospheric transport pathways, meteorological conditions along the way to Svalbard, BC ageing processes, and local meteorological processes at the glacier. All these factors contribute to what emission source areas and sectors are the most significant for the EC deposited at Holtedahlfonna, how
much of the emitted BC is scavenged and deposited from the atmosphere before reaching Holtedahlfonna, and how efficient in-cloud and below-cloud BC scavenging is at a specific time at Holtedahlfonna, and thereby how much atmospheric and in-cloud EC present at Holtedahfonna is actually deposited. These processes may vary temporally with notable effects on the observed EC deposition trend at Holtedahlfonna. As according to our model results almost 99 % of BC is wet-deposited at Holtedahlfonna, the significance of meteorological processes and their variation in comparison to
sole BC emissions for the observed EC deposition trend have to be considered. Moreover, it would be a gross oversimplification to assume that the EC deposition trend at Holtedahlfonna would solely reflect BC emission trends in source areas and/or atmospheric BC concentration trends, since local and regional meteorological processes affect the EC deposition rate notably. As a possible example of the consequences of disregarding temporal meteorological variation, previous modelling results of historical BC deposition in Finland using constant (year 1997) meteorology since 1850
show that the modelled BC deposition trend closely follows the inventory BC emission trend, while the observed BC deposition trend clearly diverged from the modelled trend (Ruppel et al., 2015). One possible explanation for the described discrepancy are variations in meteorological processes affecting BC scavenging efficiencies that were unaccounted for in the model. Thus, to produce generally more plausible modelled data, atmospheric BC deposition at Holtedahlfonna was here modelled only beginning from 1980 since when reliable meteorological data has been available.

Atmospheric BC concentration trends, on the other hand, have been generally observed to follow BC emission trends in the Arctic (e.g. Sharma et al., 2013). In our results the modelled atmospheric BC concentration decreases between 1980 and 2015 (Fig. 5̶6), as has also been observed between 1990 and 2009 at the three long-term Arctic BC monitoring stations in Alert, Barrow and Ny-Ålesund (Sharma et al., 2013), and in a 47-year weekly measurement record from northern Finland (Dutkiewicz et al., 2014). The modelled atmospheric BC concentrations at Holtedahlfonna

underestimate the values measured at the closest measurement station, Ny-Ålesund (Zeppelin), from 2001 to 2015 by an order of magnitude (cf. Sharma et al., 2013). However, the comparison between these sites should be done carefully, since the sites are located in different grid cells in the model and at different altitudes (Holtedahlfonna at 1150 m a.s.l. in comparison to Zeppelin at 440 m a.s.l.), and are therefore subjected to different wind and precipitation forcing. In addition, the monitoring station measures BC absorption which is converted to concentrations using a Mass Absorption Coefficient, whereas the model output is mass concentration. Despite the problems of comparing absolute measured and modelled atmospheric BC concentrations at these sites, they both show the same decreasing trend.

Notably, however, the modelled annual BC deposition does not clearly follow (or correlate to) the declining north of 40° N BC emissions (Fig. 3b) or modelled and measured atmospheric BC concentration trends (Fig. 56). Instead, the modelled BC deposition shows significant variation from year to year. The modelled BC deposition trend follows the wet deposition pattern at the site which varies mostly irrespective of peaks or minima in the atmospheric BC concentrations. Consequently, the modelled BC deposition seems for the most part to be driven by other parameters, for example by meteorological processes, rather than atmospheric BC concentrations. On a 35-year perspective, on the other hand,However, over the whole study period the modelled BC deposition trend is decreasing weakly, as discussed in Sect. 3.3., similar to the modelled atmospheric BC concentration trend, although the rate of the deposition decrease may is not be as evident as for the concentrations due to strong yearly variations (Fig. 56).

The modelled BC deposition at Holtedahlfonna is ca. a magnitude lower than the measured EC deposition in the ice and firn cores (Fig. 89). Similar notable under-estimations in modelled BC values compared to observations have been previously reported in the Arctic both for atmospheric snow BC concentrations (e.g. Dutkiewicz et al., 2014Forsström et al., 2013) and BC deposition (e.g. Ruppel et al., 2013, 2015). The modelled BC deposition trend at Holtedahlfonna does not show clear consistency with the observed EC deposition in the ice and firn cores, although some similarities can be observed. The notable variation in the measured ice/firn core EC deposition from one data point to the next in addition to the year to year variation in the modelled BC deposition highlights the significance of wet deposition patterns and underlying varying meteorological processes to the surface deposition trends. In addition, the modelled BC deposition trend seems to support a notable drop in BC deposition observed between the ice and firn core. The 300-year ice core recorded an average EC deposition of 18.5 mg m$^{-2}$ yr$^{-1}$ between 1980 and 2003, and the firn core an average EC deposition of 10 mg m$^{-2}$ yr$^{-1}$ between 2005 and 2015. This corresponds to a drop of 46 % in the observed EC deposition at Holtedahlfonna between the respective time periods. In the model results, the average BC deposition from 1980 to 2003 is 0.8 mg m$^{-2}$ yr$^{-1}$ and drops to 0.48 mg m$^{-2}$ yr$^{-1}$ between 2005 and 2015, which corresponds to a drop of 40 %. Thereby, the model data suggests that a significant drop in BC deposition may have occurred at Holtedahlfonna on a decadal scale, although according to the model results the drop does not seem to have happened as abruptly as indicated by the ice and firn core data. The model data do not indicate a clear peak in BC deposition around the late 1990s as recorded in the 2005 ice core, although the 5-year running average of the modelled BC deposition is temporarily lifted from 1994 to 1997. Similarly high BC deposition values are modelled for the mid-1980s but without a longer modelled time period it is unclear whether these modelled high 1980-90s values would represent similar increasing and peaking values on a decadal or centennial scale as recorded in the 2005 ice core between 1970 and 2000s (Fig. 78). Furthermore, the modelled deposition does not show an increasing trend from ca. 2005 to 2015 as indicated by the firn core measurements (Fig. 89).

Consequently, the model results support some features of the ice and firn core observations, such as higher EC deposition in the 1980s and 90s and a drop in deposition thereafter, but these variations are smoothed and lowered by the model in

comparison to the ice and firn core values (Fig. 89). Explanations for the observed variations being smoothed out in the model results could relate to the spatial and time resolution of meteorology and emissions. The spatial horizontal resolution of the ERA-Interim meteorology is $0.72° \times 0.72°$ and 3-hour time resolution. This resolution is quite crude for this study, as it smooths out the spatial and temporal distribution of meteorological variables and local climate parameters

at the glacier may not be represented accurately in the model. Also, the validation of ERA-Interim meteorological data shows that the main limitations in the Arctic are the positive biases in temperature and humidity below 850 hPA, and they do not capture low-level inversions (Dee et al., 2011). The first may influence whether precipitation is solid or liquid, changing the precipitation velocities, and the second may influence the mixing of the lower troposphere. In addition, the anthropogenic inventory emissions are available as monthly or annual emissions for only every 5 or 10 years (Fig. 3), and

the data is linearly interpolated between these data points. Thus, the scenario-based emission datasets may smooth out modelled BC variations in comparison to the ice and firn core observations, and consequently the year-to-year variations in modelled BC deposition are mainly driven by meteorology. Furthermore, the global bottom-up emission inventories are based on assumptions of emission factors (BC amount released from certain burned fuel using a given technology) and estimations of used fuel amounts (e.g. Bond et al., 2007), but recently the accuracy of the inventories on the quantity

and spatial allocation of BC emissions has been questioned particularly for the Arctic (Eckhardt et al., 2015; Huang et al., 2015; Winiger et al., 2017). Possible underestimation of anthropogenic (e.g. Stohl et al., 2013; Huang et al., 2015) and natural fire (Soares et al., 2015) BC emissions significant for the Arctic and their spatial and emission sectoral miss-allocation (Winiger et al., 2017) in the emission inventory driving the model, may partly cause the underestimations of atmospheric BC concentrations and consequentlythe lower modelled BC deposition in the model results in comparison

tocompared to the observed ice and firn core EC deposition, and may potentially affect the modelled BC deposition trend. Furthermore, the current model set-up does not include a parameterization for aerosol ageing, while models with ageing processes tend to show higher BC mass concentrations in the remote Arctic (e.g. Liu et al., 2011). The dry and wet deposition schemes of SILAM have been evaluated (Kouznetsov and Sofiev, 2012; Khan et al 2017, Sofiev et al, 2011), but currently in SILAM BC particles grow only based on relative humidity which may enhance dry deposition of

relatively large BC particles close to the sources, allowing the dispersion of only very small particles to the remote Arctic. Consequently, too little BC (in mass) may be transported and deposited annually in the Arctic in the model, especially during the Arctic Haze season (Fig. 5). However, without ageing in SILAM, the particles do not grow via condensation of soluble material during transportation, resulting in the particles being too small for dry deposition when reaching the Arctic. The lack of ageing processes may lead to an over-domination of Arctic wet-scavenging in the model as particles

are too small for dry deposition, and consequently the result of 99 % wet-deposition at Holtedahlfonna may be exacerbated.

Nonetheless, the modelled BC deposition variation suggests that the BC deposition trend may diverge from the atmospheric BC concentration trend on an annual scale (Fig. 56) which is likely explained by meteorological processes affecting for instance BC scavenging. Meanwhile, the model could be improved by including a temperature dependency

to the scavenging efficiency of BC, as Cozic et al. (2007) showed that the scavenging efficiency of BC increases significantly from temperatures of -20 (~10 % BC scavenged in mixed phase clouds) to 0 °C (60 % scavenged in liquid clouds).

**4.2. Sources contributing to modelled BC deposition and atmospheric concentrations at Holtedahlfonna**

In Ruppel et al. (2014) it was hypothesized that the observed increase in the Holtedahlfonna ice core EC deposition from 1970 to 2004 could have been partly caused by simultaneously increasing flaring emissions from north-western Russia. That area is a major source for BC in Svalbard (e.g. Hirdman et al, 2010; Stohl et al., 2013), and according to Stohl et al. (2013) flaring may contribute to 20–40 % of annual mean surface BC concentrations in Svalbard, but these emissions have been strongly under-estimated or even disregarded in emission inventories (Stohl et al., 2013; Huang et al., 2015). However, our current model results suggest a significantly lower contribution of flaring to the BC values on Holtedahlfonna between 1980 and 2015: ca. 7 % for the atmospheric concentrations and 2 % for the deposited BC (Fig. 6̶7). Only in sporadic years, such as 1982 and 2010, the flaring contribution is suggested to have increased to over 10 % of the total BC deposited. Interestingly, this modelled contribution of flaring matches well with state of the art dual-carbon isotope source apportionment measurements of atmospheric EC from Tiksi, north-eastern Russia, which suggested flaring to contribute only to 6 % of annual atmospheric EC concentrations at the site (Winiger et al., 2017). No increase in the contribution of flaring to total BC deposition is evident in our modeling data from 1980 to 2015, and even in case of possible continued underestimations of flaring in current emission inventories, the hypothesis of Ruppel et al. (2014) of flaring having partly caused the increase observed in 1970 to 2004 in the Holtedahlfonna ice core, can be rejected by the current modeling data.

As seen in Figure 6̶7 there are notable differences in the source contributions for the modelled BC deposition and atmospheric concentrations. While transport and domestic emissions appear to be the most important sources for BC deposited at Holtedahlfonna, the domestic sector seems to be the most important emission source for atmospheric BC concentrations at the glacier. This difference in the source contribution to the modelled BC deposition vs. atmospheric concentration can be explained by the difference in emission location, injection height, transport pathways, and removal of BC from the atmosphere. In the current setting of chemical transport models, such as SILAM, the physical properties of the emitted particles (type, size, hygroscopic properties) are characterized on a low description level, e.g. no aerosol dynamics, and thus no substantial difference in physical properties between the different emission sectors is present. Nevertheless, in long-term assessments of BC, the meteorology is key to determine transport pathways and scavenging of the particles from the atmosphere, and may thereby affect the differences between source contributions of modelled atmospheric and deposited BC.

For the modelled BC deposition, the contribution of domestic emissions has decreased while transport emissions have generally increased from 1980 to 2015, particularly when including shipping (Fig. 6̶a7a). The north of 40° N BC emissions from the transport sector have first increased from 1980 to ca. 2000 and then decreased (Fig. 3). A similar trend is also identifiable in the modelled source contribution of BC deposition at Holtedahlfonna, although in the 2010s the contribution of transport increases again (Fig. 6̶a7a). While this temporal evolution of emissions and modelled BC deposition from the transport sector resemble to some extent the observed EC deposition trend in the Holtedahlfonna ice and firn cores, the fraction of transport emissions to the total BC deposited at Holtedahlfonna seems based on the model data too low to solely explain the recorded ice and firn core EC deposition trends. Furthermore, the model results show that between 1980 and 2015 the contribution of natural fire emissions to both the atmospheric BC concentrations and BC deposition has increased (Fig. 6̶7), as also suggested by their increasing emissions (Fig. 3). Interestingly however, natural fires constitute 24 % of total BC emissions north of 40° N between 2010 and 2015 in the used emission data, but their contribution to the modelled atmospheric BC concentrations and BC deposition at Holtedahlfonna is significantly lower, ca. 5 % for both atmospheric composition and deposition in this time period. This may suggest that natural fire BC emissions are prone to be washed out of the atmosphere before reaching Svalbard. BC emissions from natural fires appear

mostly in spring and summer, and are the dominant source contributor in this season at Holtedahlfonna, but their contribution to annual deposition increases only seldom to notable values at the glacier.

In summary, emissions from the domestic and transport sector, followed by industry, seem to affect the BC values at Holtedahlfonna the most. None of the anthropogenic or natural fire emissions have varied independently or together in a manner that could solely explain the observed EC variation in the Holtedahlfonna ice and firn cores. Furthermore, the amount of BC emissions from individual sectors (Fig. 3) does not equal the modelled contribution of these emission sectors to the atmospheric BC concentrations or especially BC deposition at Holtedahlfonna (Fig. 67). Consequently, it seems most likely that meteorological processes affecting wet deposition patters at the glacier (and during transport) have had a stronger influence on the EC deposition trends at Holtedahlfonna than the BC emission trends.

## 5. Conclusions

According to a shallow firn core collected in spring 2015 from Holtedahlfonna glacier, Svalbard, EC concentrations and deposition have dropped to lower values in the 21$^{st}$ century after rapidly increasing values recorded from 1970 to 2004 at the glacier in a 300-year ice core (Ruppel et al., 2014). Neither the increasing trend from 1970 nor the rapid drop in EC deposition from 2003 to 2005 is supported by the Arctic atmospheric BC concentration measurement or BC emission inventory trends. A meso-to-global scale chemical transport model (SILAM) was run to investigate the difference in atmospheric BC concentration and BC deposition trends, and to evaluate BC emission sources affecting the Holtedahlfonna glacier between 1980 and 2015.

Modelling the long-term atmospheric concentrations and deposition of BC at Holtedahlfonna allowed discerning the annual variation and decadal trends for BC. As expected, the modelled atmospheric BC concentration trend corresponds to the declining BC emission trend. However, although the modelled BC deposition decreases weakly throughout the study period (1980-2015), the trend does not clearly follow BC emission or atmospheric concentration trends. Our results show that almost 99 % of BC mass is wet-deposited at Holtedahlfonna,. This number is probably exacerbated by the lack of aerosol ageing processes in the model which results, for instance, in the transported particles being too small for dry deposition in the Arctic, and consequently wet-scavenging overly dominating the deposition. Nonetheless, the results based on the current settings of SILAM which corroborates with the 85 to 90 % of BC wet-deposition generally suggested for the Arctic by Wang et al. (2011). Thus, precipitation and other meteorological factors (such as temperature and cloud phase (liquid, mixed or ice)) are crucial parameters as they drive the scavenging of BC, both on site and during the transport of BC to the Arctic. Consequently, it seems oversimplified to assume that the BC deposition trend would strictly follow its emission and/or atmospheric concentration trends.

The modelled BC deposition trend shows similarities with the observed ice and firn core EC trends with highest deposition values reached in the 1980s and 90s, and a subsequent decrease. The ice and firn core data show stronger variation and steeper fluctuations in EC deposition trends than the model. This is likely caused by key input data of the model, as the emission inventory data is based on emission scenarios that are only available for every 5 or 10 years, and the model is run with the same grid-size as the meteorology ($0.72° \times 0.72°$ horizontal resolution), which both smooth out temporal and spatial variations. The fact that the observed EC deposition trends fluctuate more than the modelled BC deposition trend suggests that the observed EC deposition trend is even less affected by BC emission trends than what the model results imply. Interestingly, the model results indicate differences in the source contribution of atmospheric and

deposited BC, with domestic BC emissions clearly contributing most to the atmospheric BC concentrations, and traffic and domestic emissions contributing equally to the deposited BC. This difference further underlines that meteorology, BC transport and chemical ageing influence atmospheric BC concentrations and BC deposition at Holtedahlfonna differently. Also, the source area location and whether the emissions are available throughout the year or are seasonal, affect how they contribute to the atmospheric concentrations and deposition at the glacier.

Notably, the recorded firn EC concentrations (from 2005 to 2015) are ~~somewhat~~ lower than the EC concentrations recorded in the first half of the 1980s in the 300-year ice core. Similarly decreasing BC concentrations were reported by Doherty et al. (2010) comparing ca. 1200 surface snow samples collected mainly between 2005 and 2009 to snow collected by Clarke and Noone (1985) in 1983 and 1984 from Arctic snow packs, including Svalbard. While Doherty et al. (2010) concluded that it was doubtful that BC in Arctic snow would have contributed to the rapid decline of Arctic sea ice observed since 1979 (e.g. AMAP, 2011), our ice and firn core results highlight that such snow measurements provide only temporal snap shots in a decadal perspective, and significant BC variation relevant for climate impact assessment may be overlooked without continuous records. In other words, only continuous long-term records can reliably show decadal trends upon which the significance of year-to-year variability can be assessed.

Observational data on Arctic EC or BC deposition are currently quite scarce and geographically restricted (mostly to Greenland and the European Arctic). Moreover, several firn/ice cores should be retrieved from same glaciers to assess the effect of local post-depositional processes and micrometeorology on the BC concentrations and deposition. The present data indicate that EC deposition at a Svalbard glacier is not solely driven by BC emission or atmospheric concentration trends, as basically all EC is wet-deposited and thereby mostly affected by precipitation and EC scavenging efficiency variations. However, it is currently unknown how widespread or pronounced such discrepancies between atmospheric BC and deposition trends generally are in the Arctic. Much further BC deposition data are required before general conclusions on the climatic implications on BC in the Arctic should be attempted, since it is specifically BC deposition on reflecting surfaces that amplifies the climate impact of BC in the Arctic compared to atmospheric BC. Furthermore, the current data suggests that Arctic BC deposition trends cannot straight-forwardly be reconstructed based on atmospheric BC concentration trends, or vice versa.

**Competing interests**

The authors declare that they have no conflict of interest.

**Acknowledgements**

We are deeply grateful for the support and funding received from the NordForsk Top-level Research Initiative Nordic Centre of Excellence CRAICC (CRyosphere-Atmosphere Interactions in a Changing arctic Climate), and the Academy of Finland projects 257903 and 296646. The field support was provided by Norwegian Polar Institute. Support for atmospheric aerosol observations at Zeppelin (Ny Ålesund) by the Swedish EPA is greatly acknowledged.

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

**Table 1.** Annual mean firn core EC concentration and deposition at Holtedahlfonna, Stake 10, from 2006 to 2014. The values present total EC concentrations and deposition averaged over the respective year, assuming that the EC deposition rate has stayed constant throughout the respective year. Dating uncertainties of the firn core increase the uncertainties of these values.

| Year | 2006 | 2007 | 2008 | 2009 | 2010 | 2011 | 2012 | 2013 | 2014 |
|---|---|---|---|---|---|---|---|---|---|
| EC $\mu gL^{-1}$ | 9,9 | 12,9 | 15,9 | 10,4 | 16,6 | 8,7 | 29,6 | 26,9 | 28,8 |
| EC mg m$^{-2}$ yr$^{-1}$ | 9,0 | 11,9 | 13,2 | 7,3 | 12,8 | 6,3 | 25,6 | 18,4 | 23,9 |

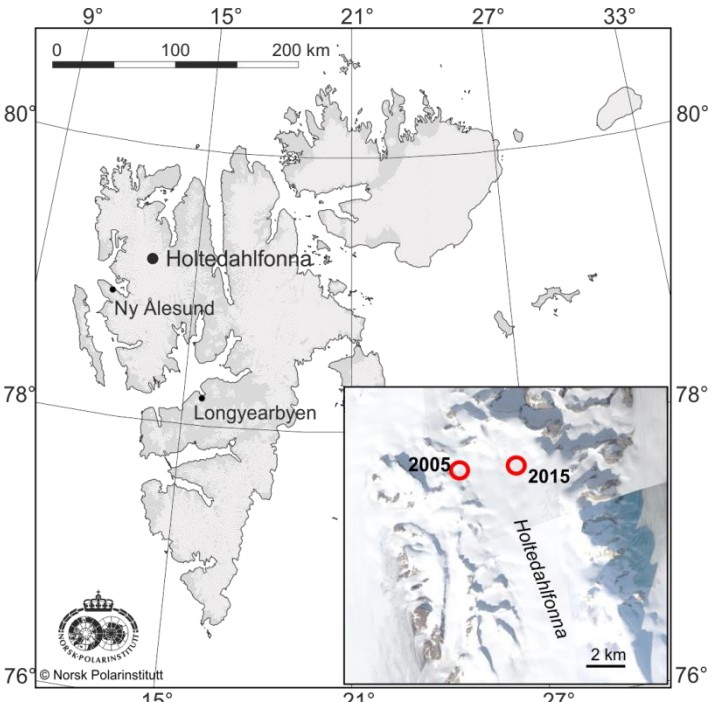

**Figure 1.** Map of Svalbard and the location of study sites on the Holtedahlfonna glacier. The inset presents an aerial satellite image of the Holtedahlfonna glacier in summer. The 300-year ice (2005) and firn (2015) core study sites are indicated by red circles.

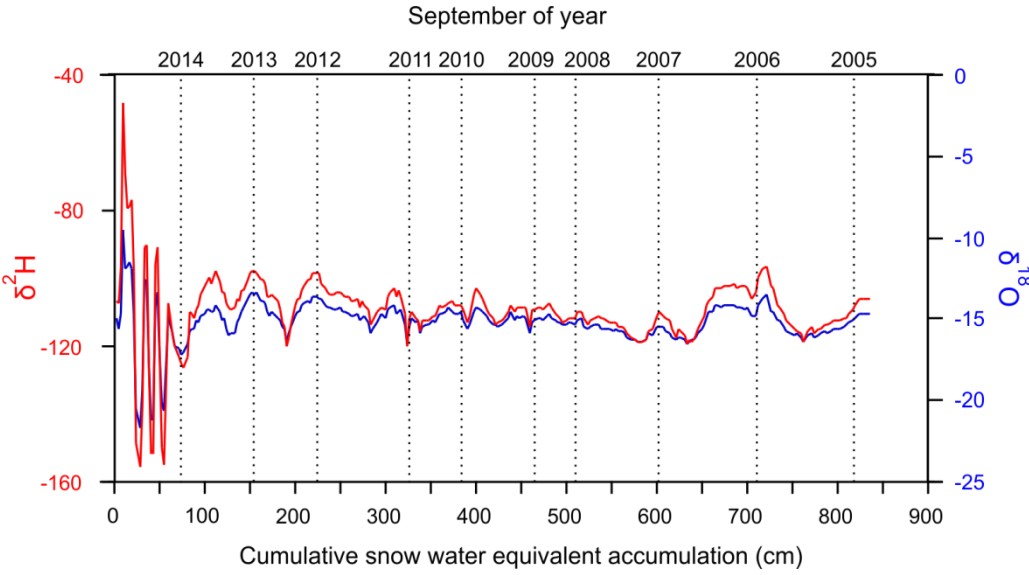

**Figure 2.** Holtedahlfonna firn core isotope and measured mass balance inferred dating. The red and blue curves present the hydrogen and oxygen isotope profiles for the firn core. The dashed lines indicate the September layers of the firn core

against the firn depth in cumulative snow water equivalent cm. This data is based on snow accumulation rate (mass balance) measurement on a nearby stake. Please see text for more details.

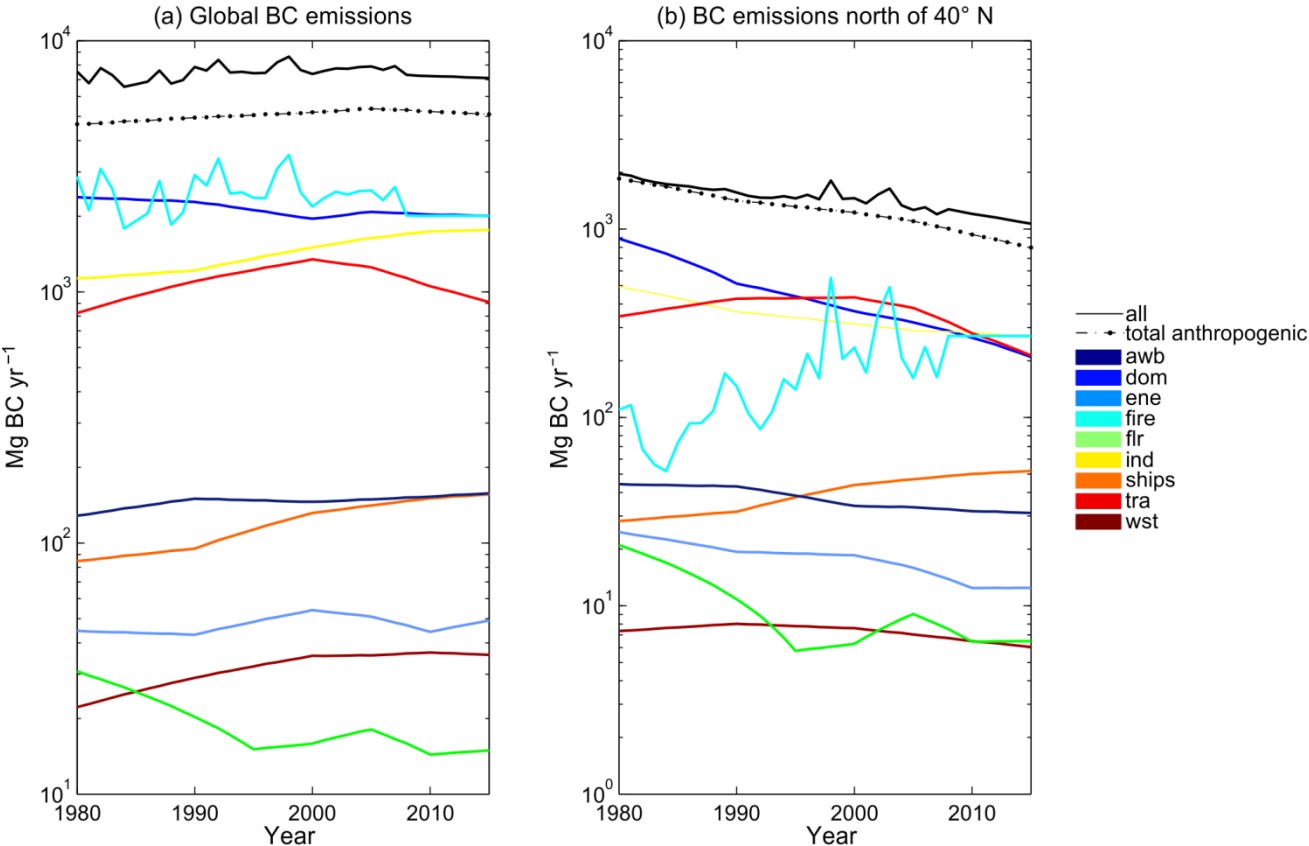

5 **Figure 3.** Temporal trend of BC emissions by sector from 1980 to 2015, (a) globally, and (b) north of 40° N. The emission sectors are as follow: all = all anthropogenic and natural sources combined, total anthropogenic = all anthropogenic sources, awb = agricultural waste burning, dom = domestic, ene = energy production, fire = natural fires, flr = flaring, ind = industry, ships = shipping, tra = transport, wst = waste incineration. Anthropogenic MACCity BC emissions are from Granier et al. (2011), ECLIPSE flaring emissions from Klimont et al. (2013), and natural fire 10 emissions from Lamarque et al. (2010). Note the logarithmic y-axes.

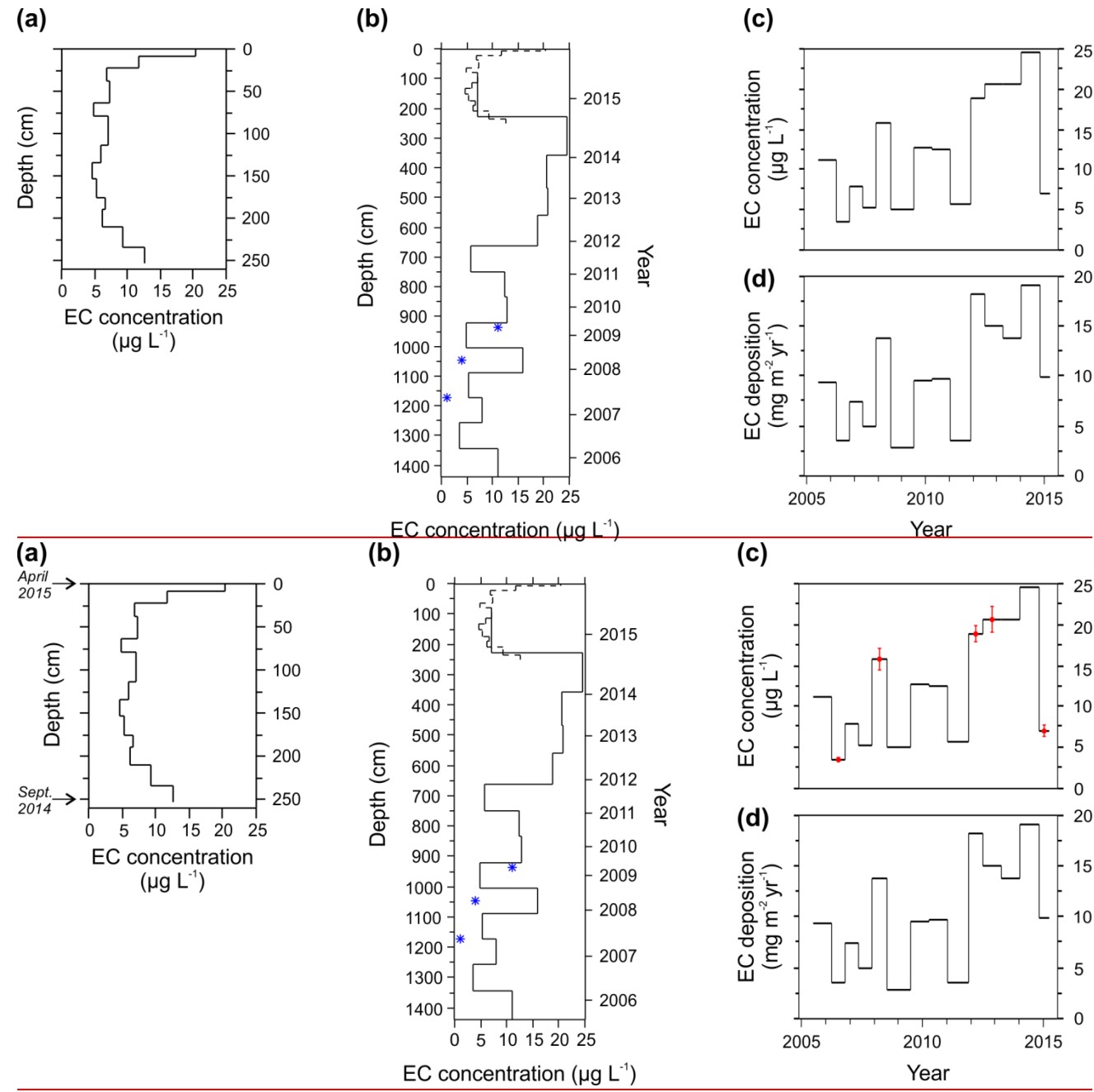

**Figure 4.** EC concentrations (µg L$^{-1}$) in the Holtedahlfonna Stake 10 snow pit and firn core, and EC deposition (mg m$^{-2}$ yr$^{-1}$). (a) EC concentrations of the snow pit against the snow depth. In addition, the April 2015 and approximate September 2014 layers are indicated. (b) EC concentrations in the firn core (*solid line*), snow pit (*dashed line*, same as (a)) and previous surface snow samples (*blue stars*) with depth and year. The previous surface snow measurements are by Forsström et al. (2013). (c) and (d) compare the temporal EC concentrations (c) and deposition (d) in the firn core. The red dots and error bars in (c) indicate average EC concentration and the absolute errors of samples from which multiple analyses were performed.

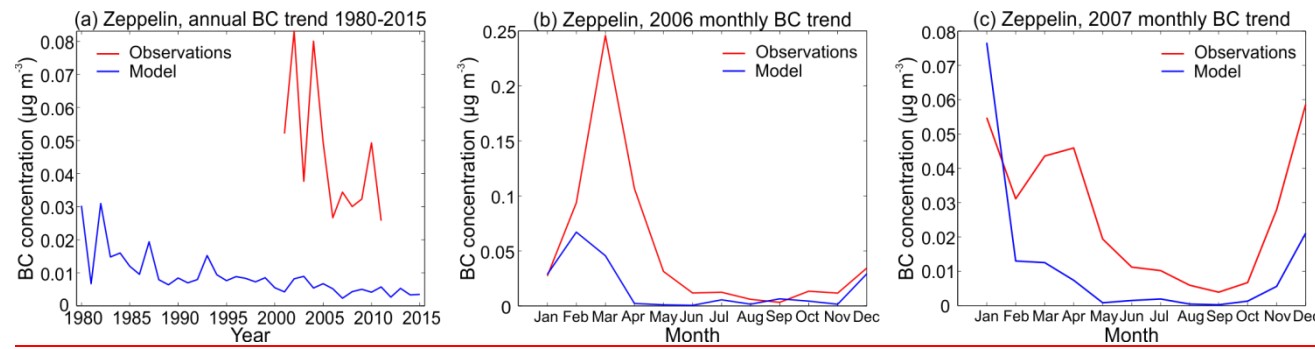

**Figure 5.** Observed atmospheric BC concentrations compared to modelled atmospheric BC concentrations at Zeppelin monitoring station Ny-Ålesund, Svalbard. (a) Modelled annual average BC concentrations from 1980 to 2015 and observed annual average BC concentrations from 2002 to 2011. (b and c) Comparison of modelled and observed monthly average atmospheric BC concentrations for 2006 (b) and 2007 (c).

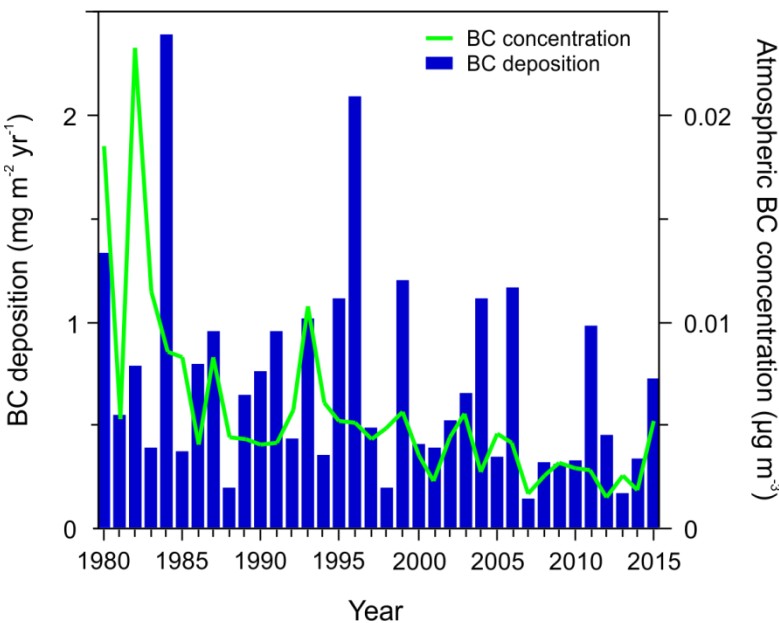

**Figure 56.** Modelled annual BC deposition and atmospheric concentrations at Holtedahlfonna.

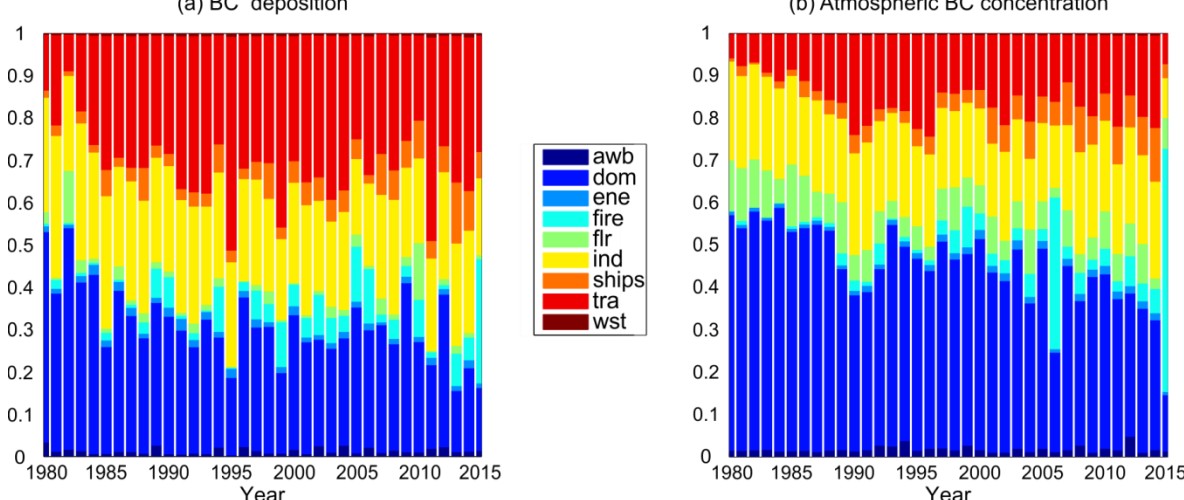

**Figure 67.** The annual source sector contribution to the modelled total BC deposition and atmospheric BC concentrations at Holtedahlfonna between 1980 and 2015. (a) The sources for BC deposition, and (b) for atmospheric BC concentrations. The emission sectors are as follow: awb = agricultural waste burning, dom = domestic, ene = energy production, fire = natural fires, flr = flaring, ind = industry, ships = shipping, tra = transport, wst = waste incineration.

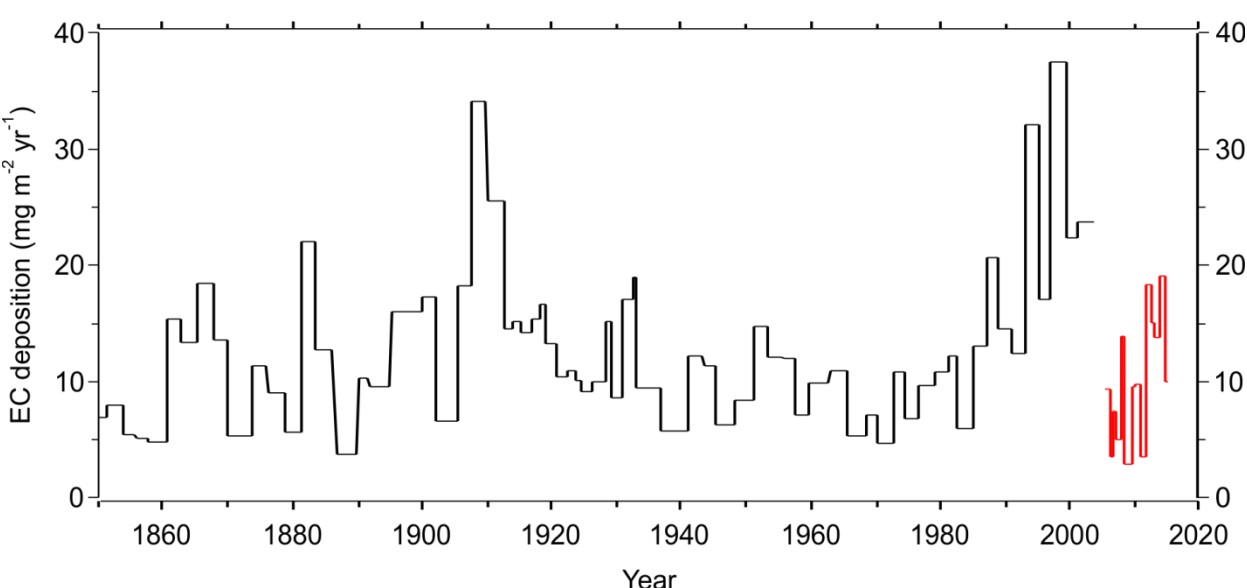

**Figure 78.** EC deposition at Holtedahlfonna between 1850 and 2015. EC deposition (mg m$^{-2}$ yr$^{-1}$) in the 2005 ice core (*black curve*, Ruppel et al., 2014) and in the shallow firn core (*red curve*) collected from different sites (see Fig. 1) on Holtedahlfonna.

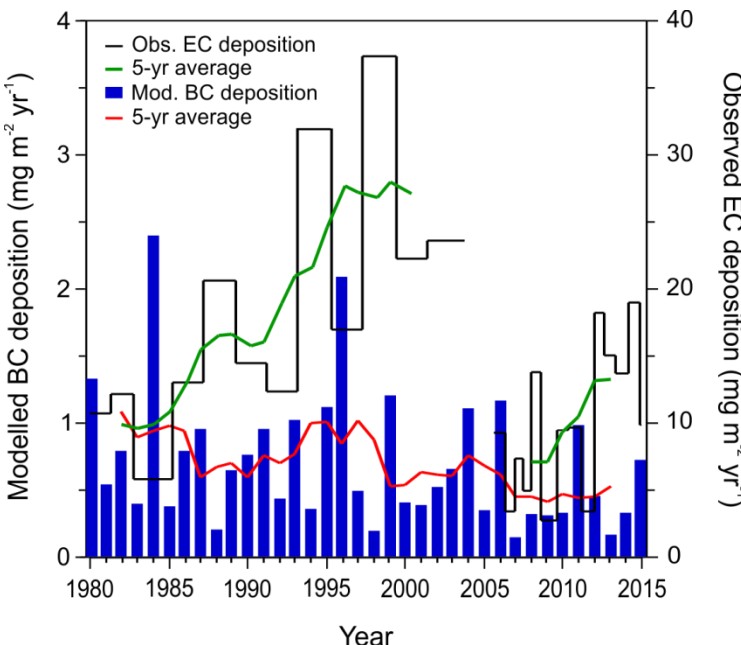

**Figure 89.** Modelled BC deposition compared to ice and firn core EC deposition at Holtedahlfonna from 1980 to 2015. 5-year running averages are included.