# Peer review of "Do contemporary (1980–2015) emissions determine the elemental carbon deposition trend at Holtedahlfonna glacier, Svalbard?"

_Atmospheric Chemistry and Physics, 2017_

## Referee Comment (RC1) · Anonymous Referee #1 · 20 Jun 2017

Review of "Do contemporary (1980-2015) emissions determine the elemental carbon deposition trend at Holtedahlfonna gloacier, Svalbard"

General comments:

I find the topic of the paper to be very interesting and timely. Increased understanding of BC deposition on snow and ice is crucial for further improvement of our understanding of climate change in the Arctic. The paper is generally well written and easy to follow. I do believe that there is lacking some discussion on the limitations of how BC is treated in the chemical transport model and, more specifically, how this affects the results and general conclusions in the work. I recommend publication if the points listed below are

[Figure]

addressed.

Specific comments:

P4, Line 14: Exchange "surface area" with "cross sectional area" for clarity.

P5, Line 29: Define Dp

P5, Line 36: Take care to specify when you are taking about atmospheric concentrations. Here it just says "modelled BC levels".

P5, Line 36: "...systematically low, but the seasonality in atmospheric BC concentrations is captured well..." Can you show this somehow? What are the observations like? Could you for example plot observed atmospheric BC concentrations at Ny Ålesund on your Figure 5. So far you have not shown or given any numbers to support your statement.

P6, Line 4: Please exchange "generally similar" with "in the same range" or something along those lines.

P6, Line 24: "... no clear decadal trend". Did you check this statistically? Visually, there seem to be a decreasing trend. You also make a statement that there is a weak trend both on p10, Line 15 and p 13 line 7.

P6, Section3.3: This section is a bit confusing. There are clear conclusions drawn here like "The model results suggest that 98.7 % of BC is wet-deposited at Holtedahlfonna." while there is no mentioning of model limitation in treatment of this or any other process for that matter. There is discussion in section 4, but at the very least, this discussion needs to be mentioned in section 3.3.

The fact that the model underestimates the atmospheric concentration by a factor of ten (?) should be emphasized and discussed in greater detail. Why does this happen? It there too much wet or dry deposition? Is the transport off? Are the emission inventories that off? Are all processes equally badly/well treated so that your statement that 98.7

% of the BC is wet-deposited holds? When 9/10 of the BC is missing (is it?) it seems a bit crude to give such a specific number for the source of surface BC?

P13, Line 9: Model limitations/uncertainty in the 99% number should be mentioned here as well.

P13, Line 19-21: Sentence starting with "The fact that the observed EC deposition trend..." How so? Please elaborate.

P13, Line 27: Consider removing "somewhat". The numbers are lower, not only briefly so.

Figure 4a): Would be nice to have time of year on the figure as you are referring to seasonality in this figure on P6 Line6.

Technical corrections:

P3, Line 25-28: Give coordinates either for both sites or none of them.

P3, Line31: Please rewrite the sentence: "To obtain a hard surface to drill 80 cm of the snow pack were removed". It is confusing.

P5, Line 35: Misspelled the word like.

P6, Line 27: Punctuation

P10, Line 21:Please insert the word "one", that is: "...deposition from one data point to the next..."

P11, Line7: "...the second may..." Correct typo.

---

## Referee Comment (RC2) · Anonymous Referee #2 · 27 Jul 2017

The study of Ruppel and others investigate how BC emitted by both natural and anthropogenic sources reaches the Arctic and deposit to the snowpack in Svalbard. The study is relevant, as it includes both measurements from snow pit and ice cores, and modeling (transport and chemistry). Transport chemical models are a great tool to investigate Arctic climate responses to emissions of short-lived pollutants. However, bringing constrains from measurements is required to evaluate such models. This paper discusses carefully multiple factors that can bring uncertainties in both BC measurements and modeling. It is easy to read, and well written. I would support its publication if the points I list below could be addressed.

[Figure]

Applying the EUSAAR_2 protocol to EC measurements in liquid phase sample is not straightforward, as eg the sample needs to be filtered, and that the efficiency of filters to capture EC can be limited (Eg Torres et al., 2014; Lim et al., 2014). Lim et al. (AMT, 2014) have reported that the filtration efficiency (ie the amount of EC retained on a filter) can be as low as 20% for small EC particle diameters (eg 100 nm MED). I am wondering if such artifact might partially explain the fact that modeled BC deposition are higher than observed EC deposition. Overall, any information about the size distributions of BC in snow/ice would be interesting, as larger particles could drive larger observed EC concentrations (and seasonal melt at the surface of the glacier can promote larger BC particles by aggregation). Considering the challenge of measuring BC in snow, combining results from different analytical methods would be more solid. If additional measurements are not possible (eg involving an SP2 analyzer) for this study, the manuscript should at least include more discussion about potential uncertainties related to the analytical method. I understand that discussion on that topic is included in the 2014 Ruppel paper, but I would recommend at least to refer more clearly to it.

The EC data should include quantified uncertainties.

The paper misses a direct comparison between model atmospheric BC and direct atmospheric observations, eg from Ny Alesund Station. This would support more clearly the model outputs (as only suggested in the manuscript).

I understand that observed EC deposition does not corroborate directly with global-scale emission patterns. Can we learn more by considering regional emissions patterns? The emissions description seems to miss details about such regional patterns, and their relative impacts at the study site.

P5-L35 : typo "llike"

P11-l7 typo "second may"

---

## Author Comment (AC1) · 1 Sep 2017

Dear Anonymous Referee #1,

We are grateful for your efforts and overall positive evaluation of our manuscript, and the constructive comments that have helped us to further improve the paper. We find all your comments well-justified and have revised the manuscript accordingly. Below we give our detailed responses to your comments and describe the revisions prepared for the manuscript. The Referee comments are cited with REFEREE 1, our responses follow in regular type, while revisions prepared to the manuscript are marked with quotation marks.

[Figure]
* * *
Interactive
comment

REFEREE 1: P4, Line 14: Exchange "surface area" with "cross sectional area" for clarity.

Thank you, done (page 4, line 14-15 of revised manuscript).

REFEREE 1: P5, Line 29: Define Dp.

Sorry, Dp means particle diameter. This has been clarified in the revised manuscript (P6, L4).

REFEREE 1: P5, Line 36: Take care to specify when you are taking about atmospheric concentrations. Here it just says "modelled BC levels".

Thank you, the statement has been clarified to "modelled BC concentrations and deposition" (P6, L16).

REFEREE 1: P5, Line 36: ": : :systematically low, but the seasonality in atmospheric BC concentrations is captured well: : :" Can you show this somehow? What are the observations like? Could you for example plot observed atmospheric BC concentrations at Ny Ålesund on your Figure 5. So far you have not shown or given any numbers to support your statement.

Thank you, we agree that it is important to show data to support our statements. Consequently, we have added two paragraphs for model validation in the model results section 3.3. and a new Figure 5, according to the suggestions of the reviewer. By adding the atmospheric concentration measured at Ny-Ålesund the reader can see that the model, though underestimating, reproduces the decreasing trend in surface concentrations (Fig. 5a). Furthermore, Figures 5b and c show that the model reproduces the seasonal pattern of EC concentration, though underestimates the size of the Arctic Haze spring peak in atmospheric BC concentrations.

The following text was added (P7, L4-19): "To evaluate the performance of the SILAM model for Svalbard, atmospheric BC observations made at the Zeppelin measurement site were compared to model results from the correspondent model grid-cell in Figure

5. Figure 5a shows the model results for the whole study period from 1980 to 2015 while atmospheric observations were available only for 2002 to 2011. Both the observations and model results show large variation in atmospheric BC concentrations from one year to the next, but with an overall decreasing trend (Fig. 5a). However, compared to the observations, the model significantly underestimates the atmospheric BC concentrations (by a factor of five on average). Such underestimations of atmospheric BC concentrations are particularly common for the Arctic where previous comparisons to observations have shown BC concentrations being underestimated in chemistry models by up to a magnitude (e.g. Koch et al., 2009; Lee et al., 2013; Dutkiewicz et al., 2014). Figure 5b and c present the seasonality of observed and modelled monthly BC concentrations in 2006 and 2007. The evaluation of the monthly model performance shows that the model captures the seasonality seen in the observations but fails to reproduce the magnitudes observed especially in spring time. Note that the timing of observed spring peaks (Arctic Haze) varies from year to year. This corroborates with several multi-model studies (Shindell et al., 2008; Koch et al., 2009; Eckhardt et al., 2015) showing that atmospheric models are usually not able to simulate the seasonality of BC in the Arctic precisely, typically underestimating the Arctic haze season occurring during the winter and early spring. A more detailed discussion on the uncertainties of the model and input driving the runs is presented in Section 4."

REFEREE 1: P6, Line 4: Please exchange "generally similar" with "in the same range" or something along those lines.

Sure, done (P6, L17).

REFEREE 1: P6, Line 24: ": : : no clear decadal trend". Did you check this statistically? Visually, there seem to be a decreasing trend. You also make a statement that there is a weak trend both on p10, Line 15 and p 13 line 7.

Thank you for pointing this out. We have clarified our statements and added the statistics of the discussed trends (P7, L21-25): "The modelled annual atmospheric BC concentrations decrease quite constantly from 1990 onwards after notably higher values modelled for the 1980s (slope -1.3 × 10-5 $\mu$g m-3 yr-1; p < 0.001). The modelled BC deposition on the other hand shows significant variation from year to year with no clear trend over the study period. Statistically, the deposition trend decreases weakly over 1980 to 2015, but this trend is not significant (slope = -3.9 × 10-3 $\mu$g m-3 yr-1; p = 0.09)."

The weakly declining trend is also clarified in the other parts of the manuscript pointed out by the reviewer.

We also added description on the regression model used for the estimation of trends in the method section (P6, L 11-14): "Subsequent to the SILAM runs, a multilinear regression model based on the median values for atmospheric BC concentrations and deposition for every single year, between 1980 and 2015, was used to estimate the slope of the modelled temporal BC trends, with coefficients being estimated with 95% confidence intervals. An F-test was applied to test if the linear regression relationship between the response and predictor variables was significant."

REFEREE 1: P6, Section3.3: This section is a bit confusing. There are clear conclusions drawn here like "The model results suggest that 98.7 % of BC is wet-deposited at Holtedahlfonna." while there is no mentioning of model limitation in treatment of this or any other process for that matter. There is discussion in section 4, but at the very least, this discussion needs to be mentioned in section 3.3. The fact that the model underestimates the atmospheric concentration by a factor of ten (?) should be emphasized and discussed in greater detail. Why does this happen? It there too much wet or dry deposition? Is the transport off? Are the emission inventories that off? Are all processes equally badly/well treated so that your statement that 98.7% of the BC is wet-deposited holds? When 9/10 of the BC is missing (is it?) it seems a bit crude to give such a specific number for the source of surface BC?

This is a well-justified comment by the reviewer. For clarity, section 3.3. presents

results from the model, and these are discussed and analyzed in section 4. Therefore, we have not included discussion of model limitations in Section 3.3. However, based on this and other comments of the reviewer, we have included some model performance evaluation in Section 3.3., as discussed above. In addition, according to the suggestion of the reviewer, we mention in the revised Sect. 3.3. that discussion on the model limitations follows in Sect. 4 (see above).

In the revised manuscript we emphasize that chemistry models very commonly underestimate BC observations in the Arctic by even a magnitude (Sect. 3.3.). In addition, we have included discussion in Section 4.1.2. on the limitations of the used model setup and why these may affect the different BC concentration and deposition magnitudes between model results and observation.

P12, L16-31: "Possible underestimation of anthropogenic (e.g. Stohl et al., 2013; Huang et al., 2015) and natural fire (Soares et al., 2015) BC emissions significant for the Arctic and their spatial and emission sectoral miss-allocation (Winiger et al., 2017) in the emission inventory driving the model, may partly cause the underestimations of atmospheric BC concentrations and consequently lower BC deposition in the model results compared to the observed ice and firn core EC deposition, and may potentially affect the modelled BC deposition trend. Furthermore, the current model set-up does not include a parameterization for aerosol ageing, while models with ageing processes tend to show higher BC mass concentrations in the remote Arctic (e.g. Liu et al., 2011). The dry and wet deposition schemes of SILAM have been evaluated (Kouznetsov and Sofiev, 2012; Khan et al 2017, Sofiev et al, 2011), but currently in SILAM BC particles grow only based on relative humidity which may enhance dry deposition of relatively large BC particles close to the sources, allowing the dispersion of only very small particles to the remote Arctic. Consequently, too little BC (in mass) may be transported and deposited annually in the Arctic in the model, especially during the Arctic Haze season (Fig. 5). However, without ageing in SILAM, the particles do not grow via condensation of soluble material during transportation, resulting in the particles being too small for

dry deposition when reaching the Arctic. The lack of ageing processes may lead to an over-domination of Arctic wet-scavenging in the model as particles are too small for dry deposition, and consequently the result of 99 % wet-deposition at Holtedahlfonna may be exacerbated."

In addition, we mention also in the conclusions of the manuscript how the limitations of the model may affect the result of the model indicating that 98.7 % of BC is wet-deposited at Holtedahlfonna (see next response).

REFEREE 1: P13, Line 9: Model limitations/uncertainty in the 99% number should be mentioned here as well.

No specific uncertainty range can be given for the 99 % number but discussion on the model limitation is added here as well (in addition to the comment above).

P14, L22-26: "Our results show that almost 99 % of BC mass is wet-deposited at Holtedahlfonna. This number is probably exacerbated by the lack of aerosol ageing processes in the model which results, for instance, in the transported particles being too small for dry deposition in the Arctic, and consequently wet-scavenging overly dominating the deposition. Nonetheless, the results based on the current settings of SILAM corroborate with the 85 to 90 % of BC wet-deposition generally suggested for the Arctic by Wang et al. (2011)."

REFEREE 1: P13, Line 19-21: Sentence starting with "The fact that the observed EC deposition trend: : :" How so? Please elaborate."

The reviewer was correct to point out that the respective sentence was unclear. We have deleted the sentence, as we thought it was a circular argument.

REFEREE 1: P13, Line 27: Consider removing "somewhat". The numbers are lower, not only briefly so."

Thanks, done.

[Figure]

REFEREE 1: Figure 4a): Would be nice to have time of year on the figure as you are referring to seasonality in this figure on P6 Line6.

In the revised Figure 4a the April 2015 layer at the surface of the snow pit and the approximate September 2014 layer at the bottom of the snow pit are indicated.

REFEREE 1: Technical corrections: P3, Line 25-28: Give coordinates either for both sites or none of them.

OK, coordinates given for both coring sites in the revised manuscript.

REFEREE 1: P3, Line31: Please rewrite the sentence: "To obtain a hard surface to drill 80 cm of the snow pack were removed". It is confusing.

Sure, the sentence reads now (P3, L31-32): "Before drilling, the top 80 cm of the snow pack were removed to obtain a hard surface to drill."

REFEREE 1: P5, Line 35: Misspelled the word like.

Thanks, corrected.

REFEREE 1: P6, Line 27: Punctuation

Thanks, corrected.

REFEREE 1: P10, Line 21:Please insert the word "one", that is: ": : :deposition from one data point to the next: : :"

Thanks, done.

REFEREE 1: P11, Line7: "...the second may: : :" Correct typo.

Thanks, done.

---

## Author Comment (AC2) · 1 Sep 2017

Dear Anonymous Referee #2,

We are grateful for your efforts and overall positive evaluation of our manuscript, and the constructive comments that have helped us to further improve the paper. We find your comments well-justified and have revised the manuscript accordingly. Below we give our detailed responses to your comments and describe the revisions prepared for the manuscript. The Referee comments are cited with REFEREE 2 and our responses in regular type while revisions prepared to the manuscript are marked with quotation marks.

[Figure]

REFEREE 2: Applying the EUSAAR_2 protocol to EC measurements in liquid phase sample is not straightforward, as eg the sample needs to be filtered, and that the efficiency of filters to capture EC can be limited (Eg Torres et al., 2014; Lim et al., 2014). Lim et al. (AMT, 2014) have reported that the filtration efficiency (ie the amount of EC retained on a filter) can be as low as 20% for small EC particle diameters (eg 100 nm MED). I am wondering if such artifact might partially explain the fact that modeled BC deposition are higher than observed EC deposition. Overall, any information about the size distributions of BC in snow/ice would be interesting, as larger particles could drive larger observed EC concentrations (and seasonal melt at the surface of the glacier can promote larger BC particles by aggregation). Considering the challenge of measuring BC in snow, combining results from different analytical methods would be more solid. If additional measurements are not possible (eg involving an SP2 analyzer) for this study, the manuscript should at least include more discussion about potential uncertainties related to the analytical method. I understand that discussion on that topic is included in the 2014 Ruppel paper, but I would recommend at least to refer more clearly to it.

The reviewer highlights important points and we agree with these uncertainties in the filter based EC measurements. Undercatchment of small EC particles is a known error source in these filter based EC measurements, causing a ca. 22 % underestimation of EC concentrations in the used set-up (Forsström et al., 2013). We agree that we should discuss this issue more, and have subsequently added this information to Section 2.2. and made clearer references to the Ruppel et al. (2014) paper where this issue has been discussed in more detail.

P4, L32-39: "The used methodology includes uncertainties that are described in more detail in Ruppel et al. (2014). In short, in liquid samples smallest EC particles may go through the filter leading to a quantified undercatchment of ca. 22 % for the used filtering set-up (Forsström et al., 2013). In addition, from each filter sample (11.34 cm-2) only a smaller punch (1.5 cm-2) is analysed for EC. To evaluate the uncertainties

caused by this subsampling, triplicate analyses were prepared for five ice core samples. These measurements (Fir. 4 c) showed an average relative standard deviation of 8.5 % (range of relative standard deviation = 5.3–13.7 %) that is smaller than reported e.g. in Ruppel et al. (2014). Combined (added together in quadrature) these error sources cause a ca. 23.6 % uncertainty in our current EC measurements."

However, the reviewer mentions by mistake that the filtrations inefficiency may potentially cause the observed EC deposition to be lower than the modelled BC deposition, while in fact it is the other way around: the observed EC deposition is significantly higher than the modelled BC deposition. The fact that observed and modelled BC concentrations and deposition very often deviate significantly from one another (e.g. Koch et al., 2009, 2011) is mostly not related to uncertainties in the observational measurements but the model parameterization (e.g. sizes of BC particles and their definition potentially deviating from what has been measured, emissions, transportation and ageing of BC particles etc.). We have added discussed model limitations and why these will affect the modelled BC values on P 12, L 16-31, also according to the suggestions of Referee #1.

We agree with the reviewer that considering the challenges in BC measurements from snow it would be most solid to combine results from different analytical methods. Unfortunately, the SP2 methodology was not available for this study but efforts are taken to make such measurements in future studies.

REFEREE 2: The EC data should include quantified uncertainties.

Yes, we agree. Quantified uncertainties have been added in Figure 4c for the ice core measurements. In addition, we have included some new text on the issue on P 4, L 32-39, as cited above.

REFEREE 2: The paper misses a direct comparison between model atmospheric BC and direct atmospheric observations, eg from Ny Alesund Station. This would support more clearly the model outputs (as only suggested in the manuscript).

Thank you, this is a justified concern that was also raised by the other referee. Accordingly, a new Figure (5) comparing observed and modelled annual average and monthly atmospheric BC concentrations at Ny-Ålesund has been added, together with two paragraphs in Section 3.3. for model validation (P7, L 4-19). For more details see responses to Referee #1.

REFEREE 2: I understand that observed EC deposition does not corroborate directly with globalscale emission patterns. Can we learn more by considering regional emissions patterns? The emissions description seems to miss details about such regional patterns, and their relative impacts at the study site.

This is a justified comment by the reviewer. We have included the emission trends of $40°$ N in the manuscript because this region is considered a significant source region for BC deposited in the Arctic (e.g. AMAP 2011). It is also shown in several back-trajectory modeling studies that northern Russia is a strong (or even dominant) source region for BC arriving in Svalbard (e.g. Hirdman et al., 2010 and references therein; Stohl et al., 2013; Winiger et al., 2015). Therefore, it could be meaningful to compare the BC concentration and deposition trends at Holtedahlfonna with Russian BC emissions. However, several studies have implicated that the BC emission inventories from Russia have not been reliable or up to date (e.g. Stohl et al., 2013; Huang et al., 2015). Huang et al. (2015) published the first regional scale emission inventory for Russia showing severe underestimation in total BC emissions, particularly from flaring, and also miss-allocations of emission sources. Furthermore, the up-dated emission inventory was published only for 2010. Consequently, it would not be meaningful to attempt any deeper comparison of regional emission patterns with our current data, particularly as the temporal trend of up-dated emission inventories would not be possible.

One of the objectives of this manuscript was to assess whether flaring or any other individual emission source could have been responsible for the EC deposition increase observed at Holtedahlfonna from 1970 to 2004. For this, it was important to use consistent emission data available for the whole study period, although it may not have

been the most up-to-date or accurate. In future, it would surely be fruitful to assess the emissions on a more regional scale but this was not in the scope of the current paper. Furthermore, our own study shows the limitations of chemical transport models to capture regional details such as the seasonal variation in BC at a remote location. This is why our study focuses on trends over several decades. Other models, such as the trajectory models (e.g. Grythe et al., 2017), would probably be able to better tackle this particular research question of how regional emission patterns affect our Svalbard study site. For clarity, we have added information on the source areas relevant for the study site in the revised manuscript, while we have not included new discussion on regional emission patterns affecting the study site due to reasons listed above.

P5, L26-27: "Generally, BC emissions north of $40°$ N are considered significant for the Arctic (AMAP, 2011)."

P5, L34-38: "The total global BC emissions have increased in the study period while north of $40°$ N they have decreased (Fig. 3). Svalbard receives atmospheric transportation dominantly from Eurasia (e.g. AMAP, 2011), and anthropogenic BC emissions from this region have decreased in the study period while natural fire emissions have increased (e.g. Bond et al., 2007; Lamarque et al., 2010)."

Clarification: P11, L8-9: "Notably, however, the modelled annual BC deposition does not clearly follow (or correlate to) the declining north of $40°$ N BC emissions (Fig. 3b) or modelled and measured atmospheric BC concentration trends (Fig. 6)."

REFEREE 2: P5-L35 : typo "llike"

Thanks, corrected.

REFEREE 2: P11-l7 typo "second may"

Thanks, corrected.

References:

AMAP: The Impact of Black Carbon on Arctic Climate (2011). Arctic Monitoring and Assessment Programme (AMAP), Oslo, Norway, 72 pp, 2011.

Forsström, S., Isaksson, E., Skeie, R. B., Ström, J., Pedersen, C. A., Hudson, S. R., Berntsen, T. K., Lihavainen, H., Godtliebsen, F., and Gerland, S.: Elemental carbon measurements in European Arctic snow packs, J. Geophys. Res. -Atmos., 118, 13614–13627, doi:10.1002/2013JD019886, 2013.

Grythe, H., Kristiansen, N. I., Zwaaftink, C. D. G., Eckhardt, S., Strom, J., Tunved, P., Krejci, R., and Stohl, A.: A new aerosol wet removal scheme for the Lagrangian particle model FLEXPART v10, Geosci. Model Dev., 10, 1447-1466, DOI: 10.5194/gmd-10-1447-2017, 2017.

Koch, D., Schulz, M., Kinne, S., McNaughton, C., Spackman, J. R., Balkanski, Y., Bauer, S., Berntsen, T., Bond, T. C., Boucher, O., Chin, M., Clarke, A., De Luca, N., Dentener, F., Diehl, T., Dubovik, O., Easter, R., Fahey, D. W., Feichter, J., Fillmore, D., Freitag, S., Ghan, S., Ginoux, P., Gong, S., Horowitz, L., Iversen, T., Kirkevåg, A., Klimont, Z., Kondo, Y., Krol, M., Liu, X., Miller, R., Montanaro, V., Moteki, N., Myhre, G., Penner, J. E., Perlwitz, J., Pitari, G., Reddy, S., Sahu, L., Sakamoto, H., Schuster, G., Schwarz, J. P., Seland, Ø., Stier, P., Takegawa, N., Takemura, T., Textor, C., van Aardenne, J. A., and Zhao, Y.: Evaluation of black carbon estimations in global aerosol models, Atmos. Chem. Phys., 9, 9001–9026, https://doi.org/10.5194/acp-9-9001-2009, 2009.

Hirdman, D., Sodemann, H., Eckhardt, S., Burkhart, J. F., Jefferson, A., Mefford, T., Quinn, P. K., Sharma, S., Ström, J., and Stohl, A.: Source identification of short-lived air pollutants in the Arctic using statistical analysis of measurement data and particle dispersion model output, Atmos. Chem. Phys., 10, 669–693, doi:10.5194/acp-10-669-2010, 2010.

Huang, K., Fu, J. S., Prikhodko, V. Y., Storey, J. M., Romanov, A., Hodson, E. L., Cresko, J., Morozova, I., Ignatieva, Y., and Cabaniss, J.: Russian anthropogenic black

carbon: Emission reconstruction and Arctic black carbon simulation, J. Geophys. Res. Atmos., 120, doi:10.1002/2015JD023358, 2015.

Koch, D., Bauer, S., Del Genio, A., Faluvegi, G., McConnell, J. R., Menon, S., Miller, R. L., Rind, D., Ruedy, R., Schmidt, G. A., Shindel, D.: Coupled aerosol-chemistry-climate twentieth century transient model investigation: Trends in short-lived species and climate responses, J. Climate, 24, 2693–2714, doi:10.1175/2011JCLI3582.1, 2011.

Ruppel, M. M., Isaksson, I., Ström, J., Beaudon, E., Svensson, J., Pedersen, C. A., and Korhola, A.: Increase in elemental carbon values between 1970 and 2004 observed in a 300-year ice core from Holtedahlfonna (Svalbard), Atmos. Chem. Phys., 14, 11447–11460, doi:10.5194/acp-14-11447-2014, 2014.

Stohl, A., Klimont, Z., Eckhardt, S., Kupiainen, K., Shevchenko, V. P., Kopeikin, V. M., and Novigatsky, A. N.: Black carbon in the Arctic: the underestimated role of gas flaring and residential combustion emissions, Atmos. Chem. Phys., 13, 8833–8855, doi:10.5194/acp-13-8833-2013, 2013.

Winiger, P, Andersson, A., Eckhardt, S., Stohl, A., Semiletov, I. P., Dudarev, O. V., Charkin, A, Shakhova, N., Klimont, Z., Heyes, C., and Gustafsson, Ö.: Siberian Arctic black carbon sources constrained by model and observation, Proc. Nat. Sci. Acad. USA, 114, E1054–E1061, doi: 10.1073/pnas.1613401114, 2017.
* * *